# An Evaluation of the Formative Experiences of Students Enrolled in Postgraduate Studies in Education: Case Study in Northern Mexico

**Brenda Imelda Boroel Cervantes** [1,*], **José Alfonso Jiménez Moreno** [2], **Salvador Ponce Ceballos** [3] **and José Sánchez Santamaría** [4]

1   Faculty of administrative and social sciences, Autonomous University of Baja California, Ensenada 21100, Mexico
2   Educational Research and Development Institute, Autonomous University of Baja California, Ensenada 21100, Mexico; jose.alfonso.jimenez.moreno@uabc.edu.mx
3   Department of Professional Development, Autonomous University of Baja California, Mexicali 21100, Mexico; ponce@uabc.edu.mx
4   Department of Pedagogy, University of Castilla la Mancha, 16071 Cuenca, Spain; Jose.SSantamaria@uclm.es
*   Correspondence: brenda.boroel@uabc.edu.mx

**Abstract:** The educational journey in postgraduate programs is linked to the actors, processes and results, setting the tone for different approaches from the perspective of characterization, development and evaluation. It is summarized in a sequential manner in four stages: entry to the program, progress within the program, and the final educational stretch, where the instructor/tutor plays an important part and obtaining the diploma or degree. The goal of this research was to evaluate, using the students' perceptions, formative experiences as a result of their academic journey in postgraduate programs within education in Northern Mexico. We have used a case study based on the focus groups technique, applied to a sample of cases comprised of students enrolled in their final educational stage. The information was analyzed using inductive data analysis. The main results were grouped into three meta categories: (1) *development of professional skills for the successful design of the intervention proposal*, which unfolded into four categories; (2) *the role of the tutor during the formative process*, consisting of four analysis categories and (3) *contributions of the teaching staff in their profession*, consisting of two categories. These trends also evidence the formative abundance in the personal, academic and social training context of the students.

**Keywords:** educational journey; postgraduate; professional development; evaluation; formative experiences

## 1. Introduction

### 1.1. Context of Postgraduate Studies in Mexico

Higher education in Mexico is characterized by a complex context involving different objectives, policies, types of institutions and forms of financing. These last decades have stood out due to a rise in demand as well as supply of a wide range of studies available to Mexican population [1]. Postgraduate studies in Mexico comprise studies in a specialty, Master's or PhD degrees, and are offered in a wide range of higher education institutions (IES). Of a total of 10,737 institutions offering this level of studies, 41% have public funding while the rest function with private funding [2]. In the 2018–2019 school year, a total of 361,267 students enrolled in higher education, of which 16% (58,520) are studying a specialty, 71% (259,698) are enrolled in a Master's degree and 14% (46,049) are attending PhD programs [3]. The wide variety currently existing at these formative levels in Mexico is characterized by multiple purposes, collegiate structures and coordination between the state, institutions and the corporate sector [4].

Postgraduate studies in Mexico are not only diverse and in high demand, but also maintain a close relationship with state policies. Since the implementation of public education policies in the 1980s, which were marked by an emphasis on economic competitiveness, postgraduate educational programs have undergone assessments of their achievements and scope, thus making evaluation an important element in the institutional planning of education at this level [5]. The role of evaluations, therefore, has been established as fundamental in regards to postgraduate programs in Mexico, alongside the necessity to attend to educational needs consistent with the high demand of this type of studies.

This demand was not only typical of Mexico; all throughout Latin America, the member states of the United Nations Educational, Scientific and Cultural Organization (UNESCO), via the Ibero-American Network for Quality Assurance in Higher Education (RIACES), began to promote the implementation of agencies dedicated to evaluating the educational level offered by universities as well as the programs themselves [6]. This intended to steer agencies' focuses towards the need for accountability throughout the Mexican states [7], and aimed to classify the solid levels of education at Bachelor's and postgraduate levels.

In addition to the high demand, there has been an intention to promote competitiveness and to the evaluation associated with educational programs, and as of 2019, higher education has been a state obligation [8]. As such, higher education institutions (IES) and research centers offering bachelor's and postgraduate degrees in Mexico will change their relationship with the Mexican State. Since the beginning of the 21st century higher education was characterized by an accentuation of the elements of demand, competitiveness and assessment, following this change an emphasis on indicator metrics as an axis for planning and orientation in the Mexican universities has been outlined [9]. Of course, this condition gains an important undertone when highlighting the obligatory nature which now characterizes it.

The onset of implementing agencies and processes of evaluating postgraduate programs (for the purpose of certification, accountability and ensuring educational solidity) was 1985. The National Council for Science and Technology (CONACYT)—a public, decentralized Mexican body specializing in articulating public policies regarding research and technological development [10], created the program to Reinforce National postgraduate studies (PFPN) with the aim of strengthening postgraduate programs. Among its strategies, it implemented awarding scholarships to students in this educational level, in order to promote exclusive dedication to study, as well as defining the criteria for assigning resources to institutions [11]. Subsequently in 2002, the PFPN became the National Quality Postgraduate Studies Program (PNPC) which establishes an evaluation framework for postgraduate programs, allowing for their classification into International Competence, Consolidated, Developing and Newly created [12]. Using this classification, various postgraduate programs throughout the country seek to legitimize their formative processes and results in the PNPC.

In parallel, at the end of the 1980s, the ANUIES (The National Association of Universities and higher education institutions) presented the document Statements and contributions of ANUIES for the modernization of higher education [13]. In this document, the IES were manifesting their desire to promote strategic planning and assessment of educational processes and results. Since 1979, the ANUIES worked with a National Coordination for the Planning of Higher Education (CONPES), and in 1989 the National Evaluation Commission (CONAEVA) was formed as one of the groups within CONPES. One of the first CONPES actions was to create Interinstitutional Committees for the Evaluation of Higher Education (CIEES). The CIEES were created to diagnose educational programs at both bachelor's and postgraduate levels. Currently, their evaluation processes are aimed at certification.

The evaluations of postgraduate programs via the PNPC have a relationship with the Mexican State, which represents certain financial benefits for the IES. Conversely, the CIEES certifications, although they do not represent the direct assignation of public resources for operating postgraduate programs (such as awarding scholarships as benefits of a

PNPC certified program), are a type of legitimization of the competitiveness of education provided [14].

In order for an educational program to be recognized in one of the aforementioned levels (Newly created, Developing, Consolidated and International Competence), a process is established (particularly within the PNPC), where the institution presents evidence of the actions implemented to guarantee the educational solidity of the program. The PNPC evaluation model consists of fifteen criteria divided into four categories, which aim to reflect the minimum acceptable requirements for a program to be incorporated into the PNPC [12]. The evaluation process starts with a self-assessment, in which the educational program describes its scope and limitations in the four categories indicated by the model. Subsequently, a revision evaluation is carried out by academic peers from different IES in the country [12]. Finally, the information of said program is analyzed and the CONACYT issues a statement regarding the degree of compliance with the evaluation criteria, issues a synthesis of relevant recommendations and observations and communicates the result decision. Currently there are 2382 postgraduate programs within the PNPC. In addition to classifying the level of consolidation of educational programs, another important element is their orientation (Table 1). According to research [12], programs can be categorized into vocational and research programs. Vocational programs are fewer than those aimed at research and, unlike the latter, they are directed towards intervention and resolution of specific ambient problems, distributed as follows:

**Table 1.** Number of postgraduate programs and categorization: vocational and research within the PNPC.

| | | | Vocational | | Research | |
|---|---|---|---|---|---|---|
| Category | Number of Programs | Percentage * | Number of Programs | Percentage * | Number of Programs | Percentage * |
| Newly created | 532 | 22% | 239 | 27% | 293 | 19% |
| Developing | 941 | 39% | 429 | 50% | 509 | 33% |
| Consolidated | 657 | 27% | 158 | 18% | 499 | 32% |
| International Competence | 252 | 10% | 31 | 3% | 221 | 14% |
| Total | 2382 | 100% | 857 | 100% | 1522 | 100% |

Source: Prepared by the author with [15] data. * Incomplete data.

It is clear that the majority of postgraduate programs certified by the CONACYT are aimed at research. Specifically, in the case of vocational programs, half of them are Developing, followed by Newly created. On the other hand, in line with the goals of this paper and regarding programs in the educational field (classified in the field of knowledge called humanities and behavioral sciences), the following Table 2, shows the state of programs in terms of the CONACYT certification:

**Table 2.** Postgraduate vocational and research programs within the PNPC in the field of humanities and behavioral sciences.

| | Vocational | | Research | |
|---|---|---|---|---|
| Category | Number of Programs | Percentage * | Number of Programs | Percentage * |
| Newly created | 41 | 31% | 55 | 20% |
| Developing | 68 | 51% | 96 | 35% |
| Consolidated | 21 | 15% | 83 | 30% |
| International competence | 2 | 1% | 34 | 12% |
| Total | 132 | 100% | 268 | 100% |

Source: Prepared by the author with [15] data. * Incomplete data.

The classification of vocational programs in the field of humanities and behavioral sciences has a similar distribution as that of programs in all fields of knowledge, where most

of them are new or in the process of consolidation—at least under the terms of the CONACYT. Although this type of programs in this field of study are few (5% of the programs accredited by the CONACYT compared to 11% of research programs in this field), it reflects the emerging interest of Mexican IES in maintaining their competitiveness and displaying a solid academic training in vocational humanities and behavioral sciences programs.

### 1.2. The Formative Journey as an Evaluation Element

The formative journey in postgraduate studies can be grouped into two categories: the first one is closely related to the quantitative indicators which provide evidence of the educational journey of the students, such as graduation rate, exam failure, exam passing, degree obtention among others [16,17]. The second is more closely associated to the qualitative aspect of the process that a student undergoes during their formative journey, such as experiences, takeaways, progress, and difficulties, among others [18–20].

Given the wide range of IES [21,22], knowing exactly how the formative process occurs generally in these institutions is a complex matter. Nonetheless, there are official norms and evaluation frameworks, such as those established by the Secretary of Public Education (SEP) [23], by the CONACYT [24,25] and as found recent research [26], which provides information regarding the way in which the process must occur in such programs, without forgetting the fact that each institution defines such characteristics in its educational models.

An important aspect to highlight is the fact that, regardless of the type of institution where the postgraduate studies are offered, the formative journey is connected to actors, processes, inputs and results, which sets the tone for different approaches from the perspective of characterization, development, evaluation among others.

Within the CONACYT PNPC, the process is characterized by a formative journey associated to a syllabus or curriculum, which we could call, according to Terigi [27], the theoretical journey. This has a duration and academic requirements, which the student must meet in order to obtain the diploma or degree. The actors of the formative process are: the student, the professor, tutor, advisor and the thesis or capstone project supervisor, as well as the set of elements offered by the institutions through their educational models, such as student mobility, modalities, specialized support services, management services, facilities, equipment among others [24,25].

With respect to the CIEES, the formative process is illustrated by a curriculum that the student progresses through. This plan, aside from the set of courses or other curricular strategies, involves a set of actors who assist the students during this process: professors, tutors, advisors and specialized support services such as educational orientation. Likewise, it includes elements referring to the formative experience (such as student mobility), and offers a series or services or support to consolidate or solve problems. The aforementioned is carried out based on the possibilities of each institution's educational and pedagogical models, considering aspects such as mobility, scholarships, infrastructure, equipment etc.

Unlike the PNPC, the CIEES clearly establishes certain services, such as psychological or psycho-pedagogical assistance and comprehensive training. The aforementioned is perhaps due to the fact that these committees also evaluate bachelor's degree programs and use the same frame of reference.

There are certain characteristics of the formative processes which are regulated at national level by the Secretary of Public Education (SEP) for postgraduate programs, and which display a certain degree of homogeneity to the existing offering. One such example are the modalities, which can be of three types: on-site, distance and mixed. Likewise, the number of credits, normally associated with a curriculum or formative itinerary and with time, is at least 45 for a vocational program and minimum of 150 for PhD programs [27]. These standards, used by private higher education institutions, are also used by most public institutions as a point of reference, and are transferred to their own standards.

Another important element that plays a relevant role in the formative process is program orientation, whether vocational or research-aimed, referenced previously. Based

on this difference, the educational journey has certain particularities, with regard to the modality, the type of teaching and advisory or the capstone project. For example, capstone projects such as portfolios or intervention projects are more related to vocational programs, whereas theses and dissertations are more aimed at research programs. Additionally, in vocational programs, progressive reference is made to working in workspaces under the supervision of an external agent located in the workspace.

Thus, the educational journey can be summarized sequentially or itinerantly as follows: (1) entry to the program; (2) progress within the program, according to the curricular structure defined in the syllabus, including the courses (as the most typical structure) or other curricular strategies defined in the institutions, such as seminars, research and intervention visits, advisory, independent studies, among others. During this process, there are countless formative experiences depending on the educational model, and, of course, on the possibilities and realities of each institution; (3) in parallel with the progress within the program or during the final educational stretch, a final project is developed, which can consist of a dissertation, thesis, portfolio, intervention proposal or other, which defines the curriculum. In this stage, the tutor, thesis or capstone project supervisor plays a dominant part in the formative process [28]; (4) finally, the final product is approved so that that subsequently, the diploma (vocational programs) and the degree (Master's and PhD programs) can be obtained.

Regardless of the type of program and the dimension of analysis of the journey, evaluation undoubtedly plays an important part, whether it is from the perspective of diagnosis, follow-up or results. The information obtained will contribute to the development of support strategies for students in all their dimensions, such as the academic, personal and professional ones. Similarly, in line with [17], studying the educational journeys of students generates relevant information for decision making in educational and institutional policies.

Even though research-based studies have focused more on the level of Bachelor's degrees [29], courses aimed at postgraduate level are usually associated with final achievement and impact evaluations. There is important evidence of improvement possibilities offered by the results of evaluations carried out during the process and not only at its conclusion [30–33].

### 1.3. Context of the Postgraduate Program in Education with a Focus on Educational Intervention

The relevance of the postgraduate program in education is based on the result of the behavior analysis of the students enrolled in this educational level in the particular entity. In the case of Baja California, a state in north-western Mexico which borders the United States of America, the number of enrolled students has increased, although proportionally less than in the rest of the country. According to the Baja California Secretary of Education (SEP-BC), the statistics in the 2018–2019 school year regarding the state number of students enrolled in postgraduate programs showed 8405 students, including the public and private sector, both in vocational, Master's and PhD programs. The Master's degree programs in the field of education include 6394 students, representing 76.07% of the total number of postgraduate students in Baja California, which provides evidence of the interest in this educational level. It is worth noting the high number of professionals in programs dedicated to teacher training for basic, medium and higher education institutions.

This postgraduate program is certified by the PNPC, and is included in the category of career guidance centered around the student, which helps develop skills in the field of formative processes and educational management, thus contributing to the professionalization of the tasks in this field. Additionally, the program focuses on basic methodological training for the preparation and development of intervention projects for the field of education. It offers the possibility to identify problems in the educational context, design, apply and assess an educational intervention proposal, in order to generate insights and give rise to recommendations and proposals to educational institutions, thus creating active professionals to improve the formative processes in the particular institution. As a



goal, it seeks to consolidate the training of professionals able to successfully influence the resolution of local, regional and national educational problems, upholding the principles of innovation, social and ethical responsibility in the fields of educational management and formative processes, with a view to transform the educational environment. The curricular structure is based on the premise of curricular flexibility, offering 15 core and elective subjects. The certification system based on the number of credits provides evidence of the implementation of an Educational Intervention Program. The duration of studies is four semesters, and the program opens for new admissions every two years.

With this said, in the process of seeking to fulfil the goals of the program, specifically to explore whether students during their educational journey have managed to consolidate their professional skills with regard to educational interventions, as well as to display a critical and proactive attitude, awareness of their surroundings and social responsibility to improve the educational processes in the specific institution, the purpose of this research project was to *evaluate, using the students' perceptions, their formative experiences as a result of their academic journey in the postgraduate program in education.* This postgraduate program therefore aims to foment conditions and processes to improve learning and to impact an educational context of heightened social vulnerability, so that students develop strategies and engagement skills to foster successful education for all.

## 2. Materials and Methods

### 2.1. Context of the Case Study

Within the UABC, the postgraduate programs offered consist of 10 specialties, 31 Master's degrees and 19 PhDs in various areas of knowledge. The Master's degree in education is offered in the field of education and humanities. The locations of the study are: the Ensenada campus of the Faculty of Administrative and Social Sciences, the Faculty of Human Sciences and the Faculty of Pedagogy and Educational Innovation in the Mexicali Campus, and the faculty of Social Sciences and Humanities in the Tijuana Campus.

### 2.2. Study Design

The project was developed in the framework of the qualitative approach of the case study design. Selection was based on the search for a perspective that allows for the application of data analysis in its context, the exploration of the interaction between individuals, without disregarding the fact that the program is the main goal, and also an approach to the experience and meaning of the postgraduate studies in practice [34]. The generalization of qualitative studies lies in the development of approaches that can be transferred to other cases; a process known as transferability.

### 2.3. Participants

With the aim of understanding from the perspective of the participants how the academic-related processes are perceived among the students and their tutors in a postgraduate program in education, in north-western Mexico, the sample consisted of 30 students distributed in two groups of 15 students each, so as to represent each campus (see Table 3). The sampling was criteria-based, and the selection of the cases in the two groups was based on their meeting the following case study selection criteria: (a) students from each of the three campuses (Mexicali, Ensenada and Tijuana), (b) enrolled in the second year (fourth semester) of the program, (c) having expressed interest in participating in the project, (d) having shown significant progress in their thesis projects according to the program schedule, and (d) having received tutorship at least three times per semester.

**Table 3.** Sample distribution per Faculty and Municipality.

| Location | Municipality | Population | Sample |
|---|---|---|---|
| Faculty of Administrative and Social Sciences | Ensenada | 11 | 6 |
| Faculty of Human Sciences | Mexicali | 16 | 8 |
| Faculty of Pedagogy and Educational Innovation | Mexicali | 18 | 7 |
| Faculty of Humanities and Social Sciences | Tijuana | 17 | 9 |
| Total | | 62 | 30 |

Source: Prepared by the author.

### 2.4. Limitations of Study

The present research is a contribution in the field of postgraduate studies in Northern Mexico. However, it has certain limitations. This is a case study, and the population size is limited, therefore the findings of the study may not be generalized beyond this university postgraduate program. It is also restricted in nature due to time constraints. Nevertheless, the students in the focus groups are the strengths of the study, as they reveal not only the genuine and true challenges faced by postgraduate students but also the hindrances posed by these challenges to the completion of their degree.

### 2.5. Technique

This research paper carried out the application of focus groups. It allows for the collection of extremely useful information, in order to reach a deep level of understanding of the opinions, perceptions and evaluations of a relevant audience regarding the practice and conditions of research development [35]. Thus, this technique establishes an atmosphere of conversation that allows us to discover the meanings that the intervening actors assign to the development of the study subject. The focus group consisted of: (a) a moderator in both groups, in charge of the research project and (b) the group of students who would potentially graduate the program. It was validated that the students met the established criteria. Among the activities carried out are the following: having obtained the proper authorization from the educational authorities for the participation of the students, the time and place of the technique was agreed in person with the latter. The participants were distributed in both groups (see Table 4). The students had the goal of the research paper explained, and their voluntary participation was requested for the day and time scheduled. Similarly, participants were made aware of the fact that the information observed during the session would be audio recorded with a view of retrieving relevant information for the research objective, and would be treated on a standard of reliability, therefore it would have no repercussions whatsoever on their educational journey. The duration of the focus group was two hours.

**Table 4.** Composition of the focus groups.

| Location | Focus Group 1 | Focus Group 2 |
|---|---|---|
| Faculty of Administrative and Social Sciences | 3 | 3 |
| Faculty of Human Sciences | 5 | 4 |
| Faculty of Pedagogy and Educational Innovation | 4 | 5 |
| Faculty of Humanities and Social Sciences | 3 | 3 |
| Total of Participants | 15 | 15 |

Source: Prepared by the author.

### 2.6. Analysis Tool

The process of developing the focus group analysis tool was carried out based on the focus group planning proposed by Krueger y Casey, [36]. Its development began with setting out the goals of information collection, corresponding to the topics included in the specific objectives of this research paper. Academic papers were reviewed, in order to prove the validity of the content of this research paper. The questions were as follows:

■ In your experience as a postgraduate student in education, with emphasis on tackling present-day educational problems, what professional skills are required in order to visualize a successful design of your intervention proposal?

■ In what way do you consider that the formative conversations with your tutor during your educational journey have contributed to your personal and academic development?

■ What are the main values that distinguish a professional in educational intervention? and

■ How do you consider that the participation of your teachers during the educational process has contributed to your personal and academic development?

*2.7. Data Analysis*

The transcription of the focus group was textual and the entirety of the group audio was taken into consideration. Regarding the data reduction technique, analysis meta categories, categories and units were established, in accordance with the project goal and the data obtained. The analysis procedure followed Mayring's proposal [37]. The analysis units were encoded and incorporated into corresponding categories. In that respect, the analysis units were grammatical units, given that we used sentences which emphasized the formative aspects. Special attention was given to recovering the statements referring to the interaction with the tutor and to the development of skills and attributes during the teaching and learning processes, as a result of their postgraduate training. In order to determine the analysis meta categories and categories, we received the support of a team of academic reviewers involved in the postgraduate program, who conducted a two-step semantic assessment of the analysis units. Initially, thirteen categories were proposed, which were subsequently reduced to ten, following the second assessment. The category reduction was conducted in order to achieve the semantic integration of the analysis units, agreed upon by the team of reviewers. Inductive content analysis was used for the purpose of associating codes with text fragments, and creating projects named "hermeneutical units" (HU) which include primary documents such as citations, codes and memos.

## 3. Results

This section presents the results obtained, describing each one of the categories constructed from the analysis of the focus groups carried out with the students. Additionally, this section provides examples of codes used and fragments of analysis units.

The findings are derived from the thematic and semantic assessment of the analysis units, which are organized into ten categories, grouped into three meta categories. These meta categories were derived from the expressions most frequently repeated and were created under the principle of supporting the categories that were mutually exclusive. Thus, they and are defined as follows:

1. Development of professional skills for the successful design of the intervention proposal, which unfolded into four categories: (a) Attitude towards Work (AT); (b) Professional Values (VP); (c) Technical Knowledge (CT) and (d) External Connections (VE).

2. The role of the tutor during the formative process, consisting of four analysis categories: (a) Motivation for Academic Improvement (MSA); Localized Learning (AS); (c) Skill Development (DH) and Dialogue and Trust (DC).

3. Contributions of the teaching staff in their profession, consisting of two categories: (a) Teaching Good Practice (PD) and Social Responsibility (RS).

With this said, in order to give an account of the students' formative process, and with it, to interpret their answers to the question, "*In your experience as a postgraduate student in education, with emphasis on tackling present-day educational problems, what professional skills are required in order to visualize a successful design of your intervention proposal*?", we obtained the meta category *professional skills*. In accordance with the literature reviewed, various theorists researching Competency-Based Education define these skills as the knowledge, attitudes and values that all students must possess, regardless of their profession. In the specific case of professional or specific skills, they are the tools and knowledge which

depend exclusively on the program attended by the student, and which must be acquired during their formative journey [38]. It was found that the students identify them through their attributes (knowledge, values and attitudes), and that they also add the aspect of external connection as an indispensable action for the implementation of intervention proposals pertinent to the context, thus responding to the real needs of educational centers. It is worth noting that, in light of their responses, we integrated the second question to the analysis process: *"What are the main values that distinguish a professional in educational intervention?"*. Below are the **unique** reflections that stood out:

> *"The lessons and takeaways that my tutor left me with were very valuable; I learned how to formulate scientific documents, the confidence to present my work at a congress and the achievement of having my first article published"*.

> *"I learned how to translate my research perspective to the educational intervention. There was a transformation from the manner in which I conceived the educational reality to how I see it after my Master's degree studies"*.

In addition, Table 5 presents the common and shared analysis units of the meta category *professional skills* in the words of the study participants.

**Table 5.** Meta category: professional skills for visualizing a successful design of your intervention proposal.

| Category | Units of Analysis |
|---|---|
| Attitude towards work [AT] | "in order for an intervention project or proposal to be successful, it requires willingness, commitment and motivation to work" [P1, P8, P17].<br>"it is essential to promote good teamwork between colleagues in different educational centers" [P3, P21]<br>"the results of the project will depend on the initiative, objectivity, teamwork, analytical and critical attitude of the person in charge" [P15, P16, P23]<br>"a student with a proactive attitude, organized, critical and open to different opinions" [P6, P17]<br>"it is necessary to develop the ability to motivate your team and to connect with the environment outside the university" [P9, P19]<br>"to be aware of the needs of the users of your project, our training should have a positive impact" [P24, P27] |
| Professional Values [VP] | "responsible and autonomous" [P1, P5, P18, P23]<br>"responsible and respectful" [P4, P9, P12, P,15]<br>"responsible, dedicated and committed to their subject" [P6, P23, P28]<br>"committed, generous and responsible towards society" [P8, P15, P21]<br>"autonomous in making decisions, free and responsible" [P10, P14, P29]<br>"fair, supportive and ethical" [P19, P22] |
| Technical Knowledge [TK] | "above all, knowing the methodology that leads to carrying out this type of proposals focusing on educational interventions"[P1, P29]<br>"it is necessary to possess diagnosis skills, specifically applying qualitative and quantitative techniques"[ P6]<br>"interactive use of the knowledge and information obtained, taking into account the proper use of technology, the ability to develop conflict management and resolution skills" [P10, P17, P22]<br>P19 "adding or acquiring facilitator/guide skills for implementing an action plan"<br>"experiential learning, diagnosis, planning and intervention"[P19, P23] |
| External Connection [EC] | "Understanding that innovation has to meet a true need of educational centers" [P3, P17]<br>"To allow the exploration of the ways in which the processes and activities within the educational center develop, from the perspective of the academic community" [P5, P21]<br>"It is necessary that the research subject to be developed come from its context, in order for the intervention to be effective" [P11, P18]<br>"One of the most important skills is the analysis of educational contexts and their challenges"<br>"connection develops within us the value of empathy and service by means of institutional connection and/or a teacher" [P16, P22, P26] |

Source: Prepared by the author.

In the words of the participants, the findings in this category indicate the contribution of the postgraduate program to the acquisition of professional skills materialized in the following main components: as added value, in the acquiring and performance of said skills they recognize the process of successful connection. We have found research projects

centered on the goals, competences and abilities of the students with a view to perform successfully in the workplace [39–41]. Apart from the findings of a study on postgraduate students by Ortega, Rendon & Ortega, [42], for the development of one's professional practice it is necessary to master a set of skills which ensure a link between the institution and the public; i.e., one's skills should be necessary to manage within the community: ability to listen, transparency, reflection, critical attitude, leadership and vocation.

Another finding is the perception of the connection between different aspects which favor the students' implementation of their proposals based on the socialization required in their research paper and in the intervention. The participants' discourse alluded to the findings of different authors [38,43] who describe a few assessments regarding academic work in postgraduate programs, for example, the dialogue and trust established in the classroom between the professor and the students, giving them a legitimate connotation of knowledge exchange and learning. They also serve to develop learning processes which promote the ability to use what was learned in the classroom in different context and areas of social education, and to apply this learned knowledge naturally to the development of creativity and innovation in practice [44].

One highlight is the importance of the experience represented by the interactions with the tutors, in order to answer the question, "*In what way do you consider that the formative conversations with your tutor during your educational journey have contributed to your personal and academic development?*".

On the one hand, the following unique study fragments stand out:

"*Assigning tutors that not only perform the role of tutor for the project tutored, but also have the empathy to listen to concerns or other things which are perhaps unrelated to the academic projects, but which help build security and self-confidence was a real success*".

"*For me, the figure of the tutor was essential; our meetings were very comforting, having someone at my side who always guided me, although, on several occasions, I myself did not know where I was headed, gave me peace*".

"*Sharing extremely enriching personal and professional experiences with my tutor has been one of the most important contributions in my postgraduate training*".

On the other hand, the common statements regarding the role of the tutor according to categories are presented in Table 6.

Regarding the contribution of the teaching staff, several educational perspectives stand out: academic, technical, practical and reflective, which play a role both inside and outside the classroom. The students also expressed the fact that their teachers' attitudes are integrated with their formative background, their teaching practices and their role as education experts. In response to the question, "*How do you consider that the participation of your teachers during the educational process has contributed to your personal and academic development?*", a few distinctive fragments from the participants' discourse are presented as follows:

"During the classes we managed to dissect the Mexican Educational System, its needs, shortcomings and situations in the professional lives of teachers and researchers. This contributed in a way to improving our practice. Lastly, the determination to avoid certain non-favorable behaviors in formative processes".

"The Master's degree in education helped me grow professionally, since in the beginning, I was having a hard time speaking publicly, whereas now, I find it gratifying to share my work, with a different perspective and confidence".

The statements are presented in Table 7.

**Table 6.** Meta category: The role of the tutor during the formative process in the postgraduate program.

| Category | Units of Analysis |
| --- | --- |
| Motivation for Academic Improvement (MSA) | being matched with a great tutor, who motivated and encouraged me to see beyond the Master's degree [P1, P17, P22] <br> my tutor always considered there was more in me to give, and in spite of the adversity, he pushed me, and he pushed me quite a lot [P2, P25] <br> my tutor also motivated me to keep moving forward and not give up [P4, P15, P23] <br> my tutor's advice contributed favorably to my personal and academic growth [P6, P16] <br> it is related to the motivation to continue studying a Master's degree and in the case of studying it abroad [P9, P19] |
| Localized Learning (AS) | The contributions my tutor left me with were enriching, the lessons and takeaways that my tutor left me with were very valuable [P1, P12] <br> I have to admit that for me it was a real challenge to meet the expectations of my supervisor, and with her support I acquired solid lessons in education [P5, P15] <br> Grounding and clearing up the ideas and information that you want to make known about your project, I learned to define my goals [P3, P23] <br> My tutor was a good guide during my journey as a student and in my intervention project, from start to finish [P12, P25] |
| Skill Development (DH) | My tutor showed me that I was lacking certain tools which I had not developed, and guided me in the process of acquiring academic writing skills [P1, P3] <br> We learned from their role how to do and how not to do things, always as a result of reflection and practice [P2, P12] <br> Deviating from the subject, and that is where the tutor comes in, who looks at your subject critically and helps you understand and formulate or explain the content [P3, P14] <br> She awoke in me the ability to analyze, readjust; my tutor's mental agility when reorganizing my ideas, finding the way and making connections and constructing an intervention proposal [P6, P17] <br> development of social skills through teamwork [P18, P22] |
| Dialogue and Trust (DC) | that trust that is also necessary in order to develop projects [P1, P22] <br> the meetings with my tutor gave me peace [P2, P17] <br> my tutor's words generate security and trust [P4, P19] <br> The dialogue allowed me to express my needs and interests and receive feedback [P5, P12, P18] <br> There was always continuous communication [P10, P14, P23] <br> Her personality stood out as open to dialogue, reflection and constructive criticism [P10, P11, P15] <br> it has helped me to make connections for future work scenarios [P17, P26] |

Source: Prepared by the author.

**Table 7.** Contributions of the teaching staff in their profession.

| Category | Units of Analysis |
| --- | --- |
| Teaching Good Practice (PD) | my tutor contributed with very valuable teachings that help me today in my professional life as a teacher [P1, P6] <br> I learned various active work methods, I value the people who strive to teach us [P2, P9, P22] <br> a different perspective on things, which gives you the tools for what you can face as a teacher [P3, P6, P13,] <br> the importance of having role models, teachers who inspire and motivate you to love what you do [P4, P15, P21] <br> I learned a lot, each one of them has characteristics worthy of admiration, regarding how to practice teaching [P5, P11, P13, P25] <br> the professors, as a consequence of reflection during our classes, managed to enrich my project and steer it in the right direction [P6, P14] <br> they prepare you to face complicated situations, which allows you to draw profound meanings from your experience [P12, P16, P21] <br> education aimed at improving the community and developing the pedagogical vision beyond the classroom [P15, P17, P27] |

**Table 7.** *Cont.*

| Category | Units of Analysis |
|---|---|
| Social Responsibility (RS) | Collaboration with the external educational context is essential for the success of the project, knowing their needs and building on them [P1, P6, P17]<br>my professors contributed to the training of proactive professionals, able to make a difference in educational centers [P2, P7, P20]<br>finding strategies to reach those I must keep informed regarding updates, and seeing how to do this, due to the proper use of these strategies [P3, P10, P13, P22]<br>The limited or lack of participation of certain professors in educational centers has also influenced my way of thinking, contributing to my questioning education [P4, P7, P11, P24]<br>being sensitive to the problems in the Mexican Educational System and taking action in that respect [P5, P15, P21]<br>placing oneself in real situations, working in the educational environment, not from my own perspective but from theirs [P6, P17, P27]<br>coming into contact with vulnerable communities and becoming involved when the cause is compatible with the project goals [P11, P19, P21, P23] |

Source: Prepared by the author.

The results allowed us to materialize the findings of different authors [39,40]: the conditions provided in postgraduate studies, such as infrastructure, professors and students, in the appropriation of knowledge of acquiring the skills to establish effective relationships between all the educational actors within the programs and in the institutions that offer such programs, is essential for consolidating academic life, unrestricted to the classroom. On the other hand, [41,42] tells us that the formative quality of postgraduate studies goes beyond the learning strategies implemented in the classroom. It depends much more on the development of transversal skills—learning to be, to do, to know and to interact in real situations of the teaching practice.

## 4. Discussion

Postgraduate training has been characterized by important transformations in the academic structure, specifically with the aim of enriching the formative journeys in higher education and postgraduate programs. The organization of actors, the diversity and complexity of the university community in the classrooms and the relationship with society are topics present on agendas worldwide. Specialized higher education still faces important challenges. For example, in Europe, various specialized and permanent education strategies have been promoted [45]. Thus, since 2014, countries such as Spain have begun a process of change in their legal framework [46], to support balance in the educational journey for Master's degree and PhD studies. In depth analysis of topics such as the organization of teaching, updating study plans, supervision of practice in social contexts, successful educational experiences and the relationship between professors and students are a few of the relevant lines of research [47]. In Latin America, the educational opportunities for postgraduate studies are monitored by the International Institute for Higher Education in Latin America and the Caribbean (IESALC), which supports the necessity to possess more effective mechanisms to ensure quality in student training, so that the alumni can successfully face future obstacles and challenges [48].

Envisioning postgraduate education as education centered on the students as objects of comprehensive education, involves reflecting on the reality and the problems it faces. Listening to their voices means accepting that some of them express challenges and formative skills that are not resolved during their educational journey. They pose the need to respond from a different mindset, in order to explore the nature of the pedagogical and social needs that go beyond the educational intervention classes offered in postgraduate programs.

In the words of the participants, the approach of the educational intervention promoted in postgraduate programs is a strategy that works as a concrete resource, because it helps resolve specific problems during the formative journey. Nevertheless, it should not be considered a means in and of itself, and from their perspective, education goes beyond

content for content's sake. It means the ability to develop thinking skills, reflection and acting within the immediate reality through educational practices led by their tutors and professors, and especially, within a framework of ethics and social responsibility.

In addition, it must be acknowledged that postgraduate training takes place in multiple settings, inside and outside institutions. This helps face the challenge of establishing pedagogical approaches from the perspective of comprehensive training (knowledge, abilities attitudes and values), by promoting learning settings for the responsible and committed comprehension and experimentation of research. In that sense, the role of the tutors is essential. During tutorship, they offer elements that facilitate decision making tor continued improvement of education [49].

Finally, regarding the contribution of the teaching staff, two educational perspectives stand out: practical and reflective, which play a role inside and outside the classroom, creating a culture which supports collective endeavors and promotes cooperative and collaborative work. The results allowed us to materialize the findings of [49]: the conditions provided in postgraduate studies, such as infrastructure, professors and students, in the appropriation of the knowledge of acquiring the skills to establish effective relationships between all the educational actors within the programs and in the institutions that offer such programs, is essential for consolidating academic life, unrestricted to the classroom.

## 5. Conclusions

In the framework of the national goals for the development of postgraduate programs in Mexico, exhaustively referenced in various guidelines of educational policy, the following stand out as paramount indicators: quality, equity and pertinence. Conversely, according to [41], there is an important number of research papers on the operation of postgraduate programs (efficiency, sufficiency and quality). Nonetheless, there are still pending research avenues such as the impact of technology, the connection and the good use of material and physical assets in the quality of student training, according to the parameters of the CONACYT.

The challenges of formative processes used in postgraduate programs lie in the incorporation of the Competency-Based Education model in their education, and with it, elevating the educational quality must be based on guaranteeing comprehensive training of the students [50]. It is also important to mention that among the contributions of this educational model to comprehensive training, there is the recognition of the importance of mentoring, value training, contextualized education and the development of skills and abilities necessary in day-to-day life [38,51,52].

We must consider an education guided by sustainability as a framework for ensuring experiences that can have a positive impact on the development of a whole person's capabilities, including lifelong learning from the capabilities perspective. As a consequence, every educational process must strive to have an impact in their context; this will promote sustainable development. This is combined in this research with the intention of obtaining evidence to gain a better understanding of the implications of an educational process in postgraduate teacher education. As noted in other work [53]: "A quality education inspired by the social value of sustainability can be defined as one that is able to provide meaningful and relevant learning environments, processes and tools for all learners, ensuring access, promoting retention and contributing to educational success for all in line with SDG 4 of the 2030 Agenda framework: ensuring inclusive and equitable quality education and promoting lifelong learning opportunities for all [54]". In addition, as previously stated, this research aims to create conditions and processes to improve learning in an educational context of heightened social vulnerability, so that participants may develop strategies and engagement skills to foster successful education for all.

In this specific case, the postgraduate program under study corresponds to the characteristics of the on-site formative processes regulated nationally by the Secretary of Public Education (SEP) and of the vocational formative processes for postgraduate programs in line with the CONACYT. According to 60% of the students, the goals of the program are

met, and interesting categories stand out which provide evidence of the educational quality, as expressed in three trends: (1) development of professional skills for the successful design of the intervention proposal, which unfolded into four categories, (2) the role of the tutor in the formative process, consisting of four analysis categories, and (3) contributions of the teaching staff in their profession, consisting of two categories. These trends also show evidence of the formative abundance in the personal, academic and social training contexts of students.

The research results indicate that the achievements of the mentoring process in the postgraduate program do not fall short of those of the institutional student support program at Bachelor's degree level, nor of those in higher education, which, even though particular differences stand out, involve different aspects with the personal (social and emotional) and academic (learning) context. Furthermore, the results present elements related to the practice of the tutor within the educational center, as well as the social construction of the issue regarding the limited ability of the tutor to approach social and emotional aspects and individual needs within the student community. The findings also evidence the need to understand the types of mentoring which develop during student training, as well as to clearly establish the functions of the academic student support in developing their formative journey and in developing their capstone project and field intervention, avoiding confusion between these mentoring roles. It is necessary to continue researching by means of in-situ diagnosis, which gives students a voice, and thus to explore the subjective aspects of mentoring by identifying possible intervention areas with a view to improve the social and emotional processes related to the closeness and trust between tutors and students [55,56].

**Author Contributions:** B.I.B.C., Methodology, content analysis and draft preparation to submit to the journal; J.A.J.M., Conceptualization and supervision analysis; S.P.C., Conceptualization, and Writing: J.S.S., resources and visualization. All authors have read and agreed to the published version of the manuscript.

**Funding:** This research received no external funding.

**Institutional Review Board Statement:** The study was conducted according to the guidelines of the Declaration of Helsinki, and approved by the Ethics Committee of Postgraduate Education in the Faculty of administrative and social sciences, Autonomous University of Baja California research protocol, approved on date 15 January 2021.

**Informed Consent Statement:** Informed consent was obtained from all subjects involved in the study.

**Data Availability Statement:** The data presented in this study are available on request from the corresponding author. The data are not publicly available due to maintain confidentiality of the participants.

**Acknowledgments:** We are grateful to the Autonomous University of Baja California for the support during the research project at the postgraduate level.

**Conflicts of Interest:** The authors declare no conflict of interest.

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
