# Peer review of "An Evaluation of the Formative Experiences of Students Enrolled in Postgraduate Studies in Education: Case Study in Northern Mexico"

_sustainability, doi:10.3390/su13094790_

Round 1

Reviewer 1 Report

Perhaps, the number of students participating in focus groups per degree is not very representative, and can be improved in future studies. In addition, focus group participants could be selected by some key characteristics (based on context and background indicators).

The list of references contains errors (lines 561 and 563).

Author Response

THANK YOU FOR YOUR SUGGESTIONS FOR IMPROVEMENT OF THE ARTICLE. 
THE ARTICLE REPORTS THE RESULTS OF THE FOCUS GROUPS. THE FINDINGS WILL BE COMPLEMENTED, THE RESULTS CORRESPOND TO A LARGER PROJECT. 

CORRECTIONS TO THE REFERENCES HAVE BEEN TAKEN CARE OF ON LINE 561 AND 564.

THE LINES HAVE BEEN ADJUSTED TO 617 AND 618.

  1. Fresan, M. La c onstrucción de tipologías de Instituciones de Educación Superior en México. En G. Álvarez (Coord.), La ANUIES y la construcción de políticas de educación superior 1950-2015, México: ANUIES. 2015, 513-531.
  2. Mendoza, J. Ampliación de la oferta de educación superior en México y creación de instituciones públicas en el periodo 2001-2012. Revista iberoamericana de educación superior2015, 6, 3-32.

Reviewer 2 Report

Thank you for the opportunity to review your manuscript. Please find my specific comments below:

Abstract

When introducing the three trends, also including the number of categories for each makes it hard to read and interpret. Listing all three trends sequentially would be more reader friendly.

Introduction

The introduction is very long and quite detailed, at times making it hard for the reader to follow. I suggest creating sub-headings for various chunks of information/sections. (e.g., Background and Socio-Historical Context, The Formative Process, The Present Study, etc. )

It seems as though Tables 1-3 could be consolidated into one table.

Methods

Please provide more detail on the case study design in general (referencing research).

Does the sample consist of all students who met the inclusion criteria? Or were a subset of students chosen? If so, how were they selected?

The narrative says the participants are students in their 4th year of study, but both tables are labeled as faculty. Please fix or clarify this for the reader.

How long were the focus groups?

Data Analysis

More detail on the data analysis process is warranted. How many coders were there? Was consensus reached in some fashion? Did you measure reliability of coding?

Results

At the beginning of each subsection of the results the authors start by providing quotes from students that stand out. This is followed by a table of selected quotes from each category. As a reader, I want to know what the common sentiments were across participants (these would be the themes) and what the unique, and possibly important, sentiments are worth noting. As written, it is unclear what types of quotes the ones that stand out are, and I would be apprehensive that they were selected by the authors to make a point versus to truly represent an important perspective (in which case more context should be provided).

The analysis of the themes in the paragraphs following the provided quotes lack detail. Providing a written narrative explanation of each theme before introducing the quotes would help the reader better understand their meaning.

Discussion

It seems as though the true value of the research project gets lost when such a large emphasis is placed on the national goals and policies. At heart, the paper is a qualitative analysis of student perspectives. Why is it so important to capture their voices? What do their perspectives tell us about the current policies and ways of doing things? What can we learn from them that can inform the way universities are evaluated?

Author Response

Thank you for your suggestions and observations for the improvement of the article, they were very valuable to achieve a better writing.

The adjustments are in the attached file

Reviewer 3 Report

The title of the manuscript is to be concise, specific and relevant. The title is informative; allows the reader to assess the relevance of the article and contains terms suitable for indexing and searching.

The abstract consists of 201 words, it is a single paragraph.

The main question was posed in a broader context and the purpose of the study was highlighted

There is a short description of the main methods applied.

It describes who makes up the sample, but not how many respondents were included.

Authors summarize the article's main findings in the abstract. They indicate the main conclusions or interpretations.

It is good to give pedagogical implications regarding the obtained research results

Keywords are suitable for indexing and searching.

The article has an empirical character and is organized according to the IMRAD scheme (Introduction, Methods, Results, and Discussion).

Explanation and presentation of results and conclusions are good. Analysis and interpretation of research results are clear and systematic.

I think that the research sample is too small. Some researchers say between 30-50 sample size is fine, while others say between 5-25. There is even a renowned authority who claimed that one participant is enough depending on what a researcher is researching, but, for this kind of research the number of participants (18 students, two or three per institution) is not enough for valid generalization and conclusions.  

References

There are enough references included; references are relevant to the discipline and problem under consideration. Many references are not in English.

Line 525  27 27-38.  Missing comma after number 27

Line 563 págs. 513-531. págs should be deleted

Line 579 págs. 579 págs should be deleted

Line 583  págs. 1-175. págs should be deleted

Line 584  págs. 11-234    págs should be deleted

Line   592 188-191.   Numbers should not be bolded

Line   597  2012 30, 2, 127-152. Missing comma after number 2012

Line 561, 562, 563  Wrong number  of reference

Author Response

Thank you for your suggestions and observations for the improvement of the article, they were very valuable to achieve a better writing.

The corrections can be found in the attached file
